# Prevalence of emergency caesarean delivery and its predictors among women who give birth in Ethiopia using further analysis of EDHS 2016 data: A mixed effect model

**Muluken Chanie Agimas** 🄳 *

Department of Epidemiology and Biostatistics, Institute of Public Health, College of Medicine and Health Sciences, University of Gondar, Gondar, Ethiopia

* mulukensrc12@gmail.com

## Abstract

### Introduction

Emergency caesarean delivery is a surgical procedure that is decided after the labour pain has started. According to the global report of the World Health Organization, the rate of caesarean sections has risen over time. In Ethiopia, the overall caesarean delivery was 18%, which varied between 46% in the private sector and 15% in the public sector. But specifically, the magnitude of emergency caesarean delivery and its predictors are not well addressed in Ethiopia. Therefore, this study was aimed at assessing the prevalence and predictors of emergency caesarean delivery in Ethiopia using EDHS 2016.

### Method

A cross-sectional study was used, and a total of 11,022 samples were included in this study. The Ethiopian Demographic Health Survey 2016 data set was used as a data source. The STATA version 17 software was used for descriptive, bi-variable, and multivariable analysis. Multilevel binary logistic regression was used to identify the significant factors at a p-value of <0.05 and a 95% confidence level. Model comparison and goodness of fit was assessed by AIC.

### Results

The prevalence of emergency caesarean deliveries in Ethiopia was 1.2% (95% CI: 0.58, 1.78). History of fistula (AOR = 7.82, 95% CI: 1.59–38.4), age ≥ 35 years (AOR = 6.98, 95% CI: 3.33–14.63), and rural residence (AOR = 2.23, 95% CI: 1.25–3.21) were the predictors of emergency caesarean delivery.

### Conclusion

As compared to the previous study, the prevalence of emergency caesarean delivery was low. Women with a history of fistula, from rural residence, and age≥ 35 years were at risk for emergency caesarean delivery. Therefore, interventions need to be encouraged to give

**Data Availability Statement:** Underlying data is restricted due to ethical reasons by the The Demographic and Health Surveys (DHS) Program. To access the datasets, researchers can login at

([https://www.dhsprogram.com/data/dataset_admin/login_main.cfm](https://www.dhsprogram.com/data/dataset_admin/login_main.cfm).) or they can contact the DHS at [archive@dhsprogram.com](mailto:archive@dhsprogram.com).

**Funding:** The author(s) received no specific funding for this work.

**Competing interests:** The authors have declared that no competing interests exist.

**Abbreviations:** ANC, Antenatal Care; AOR, Adjusted Odds Ratio; AIC, Akaike Information Criteria; CI, Confidence Interval; EDHS, Ethiopian Demographic Health Survey; ICC, Intraclass correlation coefficient; PCV, Proportional Change Variance.

attention to rural women whose age is $\geq$ 35 years and fistula reduction activities, such as avoiding early marriage, to reverse the problem. Early and accurate screening of women for emergency cesarean delivery by encouraging co-services like ANC is also recommended.

## Introduction

Caesarean delivery is a surgical procedure that is performed when a life-threatening condition for the woman and fetus occurs [1]. Based on the degree of urgency, elective and emergency caesarean delivery are the most common classifications of caesarean delivery [2, 3].

Emergency caesarean delivery is a surgical procedure that is decided and performed after the labour pain has started. Whereas elective caesarean delivery is a surgical procedure decided and performed before the onset of labour pain [4]. According to the World Health Organization's (WHO) recommendation, the caesarean delivery should be between and 15% [5]. But according to the global report of the WHO, caesarean delivery has risen over time, which ranges from 7–21% from 1990–2021 [6]. In Saharan African countries, 8.8% of deliveries are through caesarean delivery [7]. In Sub-Saharan African countries, the prevalence of emergency caesarean delivery is 4.6%, which is higher than elective caesarean livery (2.7%) [8]. In Ethiopia, the overall caesarean delivery was 18%, which varied between 46% in the private sector and 15% in the public sector [9]. But specifically, the magnitude of emergency caesarean delivery and its predictors are not well addressed in Ethiopia. In developing countries, including Ethiopia, lack of knowledge about childbirth-related complications, delays in referrals, and resource-limited health care systems, including the low access to emergency caesarean delivery, are highly correlated with maternal and fetal health [10]. In Ethiopia, access to emergency caesarean delivery is available only at hospitals and specialty clinics [11, 12], and still the birth-related complications and deaths are high [9]. Because of prolonged labour, dysfunctional labour, and the high risk of induction of labour, women with advanced maternal age (>35 years old), inadequate antenatal care visits, and being nulliparous were at risk for emergency caesarean delivery [13–17]. In Ethiopia, the early neonatal adverse event after emergency caesarean delivery is 26.7% [18]. Timely screening and providing life-saving procedures for those at risk for emergency caesareans are very important to reduce maternal and perinatal mortality. But, in Ethiopia, there is no established protocol for health professionals to identify pregnant women early for emergency caesarean delivery and prompt management. This is because there is a lack of available evidence about emergency caesarean delivery. The numerous previous studies conducted in Ethiopia have considered women who have elected themselves for a planned caesarean delivery with non-medical indications and overall caesarean delivery. Particularly, studies done on emergency caesarean delivery are rare in Ethiopia. Conducting an emergency caesarean delivery is a very important tool for screening practices and developing working guidelines for health professionals. Therefore, this study was aimed at assessing the prevalence and predictors of emergency caesarean delivery in Ethiopia using further analysis of the 2016 Ethiopian demographic health survey data set.

## Methods

### Study design and setting

A community-based cross-sectional study design was employed to assess the prevalence of emergency caesarean delivery and its predictors among women who give birth in Ethiopia

using the data set of the 2016 Ethiopian demographic health survey (EDHS). Ethiopia is one of the east African countries and currently includes 12 regions such as Afar, Oromia, Amhara, south Ethiopia, south-west Ethiopia, central Ethiopia, Harari, Tigray, Somalia, Benishangul Gumuz, Gambela, and Sidama. It also has two city administrations, namely Addis Ababa and Dire Dawa. On July 25, 2023, the Gregorian calendar worldometre report showed that the total population of Ethiopia was 126,738,662, and the urban population contributes about 22.1% of the total population [19].

## Population

The source population was all reproductive-age women who gave birth within the 5 years before the survey, and the study population was all live births in the enumeration area. All women who had a history of live birth during the survey were included in the study, and women who were seriously ill during the data collection period were excluded.

## Outcome of interest

Emergency cesarean delivery (yes, no).

## Operational definition

**Emergency cesarean delivery.** A surgical delivery is decided and performed after the onset of labour pains starts and an urgent life-threatening condition for the woman and fetus occurs [2–4].

　　**Community educational status.** was also generated based upon the proportion of educated community attainment in the community as a high proportion and a low proportion [20].

　　**Community media exposure.** was produced based on the individual response to exposure. to radio and TV in each cluster as a high proportion and a low proportion of media exposure [20].

　　**Clustering variable.** EDHS cluster (V001).

## Sampling procedure and sampling technique

To select the eligible participants in the survey of EDHS 2016, a total of 645 enumeration areas, 202 and 443 enumeration areas from urban and rural residence, respectively, were selected. Each enumeration area had 28 households. These enumeration areas have a total of 18,060 households. For urban and rural total households, proportional allocation to size was used to be more representative. A multi-stage sampling technique was used to select the eligible participants. Finally, a total of 11,022 women who gave birth prior to the 2016 survey were included for analysis (**Fig 1**).

## Data collection procedures and data quality assurance

We used the previously collected EDHS 2016 data. The EDHS 2016 data were collected by 132 health professional interviewers: 66 as biomarker technicians, 33 as field editors, and 33 as team supervisors, from January 18, 2016, to June 27, 2016. To use this data for the current study, initially the data was requested online by explaining the purpose of the data request from the International Demographic Health Survey. After two consecutive working days, the data was released at the Demographic Health Survey (DHS) international official website www.measuredhs.com. In the survey, a structured interview-administered questionnaire was used to collect the data from the participants. To achieve the quality of the data, pretesting, data collection, and supervisor training were conducted [21]. After the approval of the data

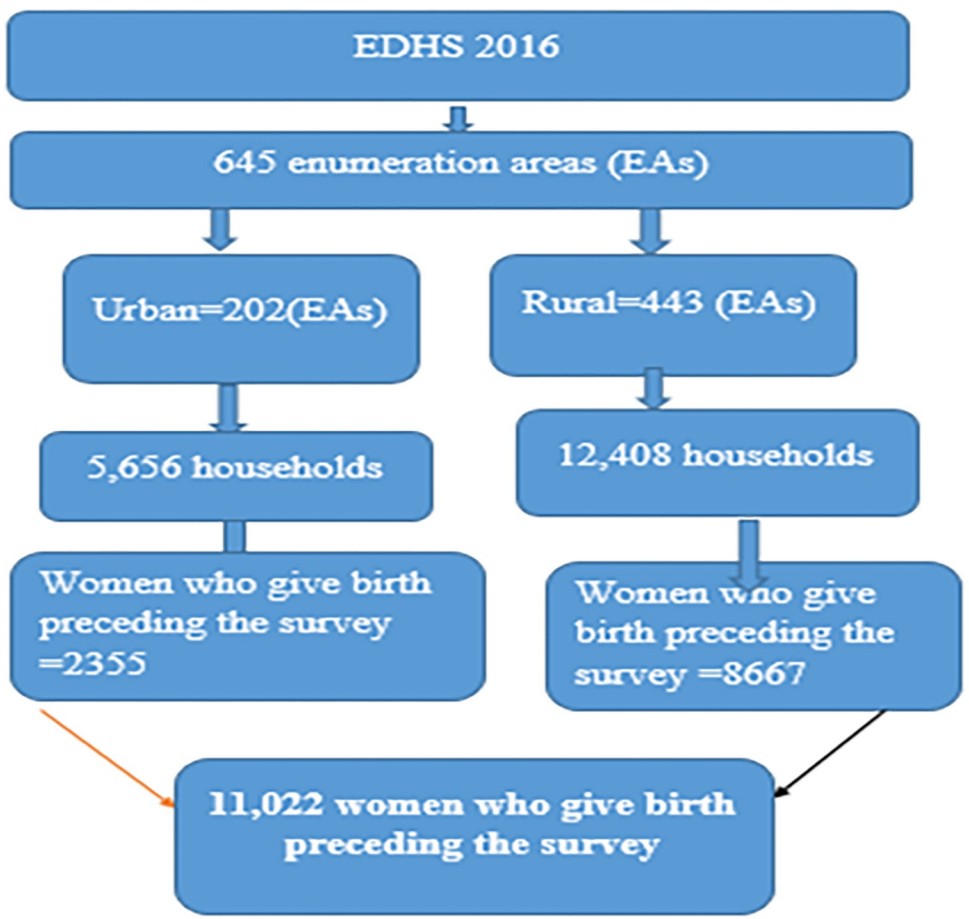

**Fig 1. Schematic diagram for the study of emergency cesarean delivery and its determinants in Ethiopia using EDHS-2016 (from 18 January to 27 June 2016).**

access, data cleaning, data recording, data extraction, and data computing were conducted to prepare for further analysis.

## Data processing and statistical analysis

After the data were accessed, STATA software version 17 was used for data cleaning, recording, computing, transforming, and multilevel binary logistic regression analysis. Descriptive analysis was also conducted using frequency and percentage. To minimize the unequal probability of selection for each strata, sampling weight was used using the "svy" STATA command. The reason we used the multilevel binary logistic regression model was because of the hierarchical nature of the EDHS data and the possibility of considering a natural nesting of data. A total of four models, such as the null model (a model without predictors), model-I (a model with level I or individual level predictors), model-II (a model with level-II or community level predictors), and model-III (a model with both level I and II predictors or a mixed model), were developed one by one. To test the clustering effect, the intra-class correlation coefficient was used with a cutoff of >0.05 (>5%). The clustering variable used to show the clustering effect of emergency caesarean delivery was the EDHS cluster (V001). For each model, Intra-class correlation (ICC ($\rho$) = $\sigma2 \epsilon/\sigma2 \epsilon + \sigma2 \mu$) was calculated [22]. The ICC of the null model (a model without a predictor) was 0.234 (23.4%). Additionally, for each developed model, a

**Table 1. A model compassion for the emergency cesarean delivery in Ethiopia using EDHS 2016 data (from 18 January to 27 June 2016).**

| Random effect | Null model | Model I | Model II | Model III |
|---|---|---|---|---|
| Variance | 0.48 | 0.13 | 0.3 | 0.1 |
| ICC | 23.4% | 6.8% | 12.2% | 5.8% |
| PCV (%) | Reference | 72.9% | 37.5% | 79.2% |
| Log likelihood | -806 | -338 | -757 | -336 |
| AIC | 1616 | 698 | 1524 | 695 |

proportional change in variance (PCV) was calculated. PCV = (variance of the null model -variance of the next model)/*100 [22].

Variance in the null model

The log-likelihood ratio test and the Akaike information criteria for each model were also calculated. The AIC is 2k-2lnL, where k is the number of parameters and L is the maximum value of the likelihood function of the model. Then the best model was selected based on the lowest AIC value. The successive decrement of the AIC value in the models implies that the model best fits the data. Based on the lowest AIC criteria, model III (the mixed effect model) was the best model. This final model had an AIC goodness of fit value of 695 and a PCV of 79.2%. The details of the AIC, ICC, LLR, and variance of each model are reported in (**Table 1**).

## Predictors of emergency cesarean delivery

The possible candidate variables were selected by bi-variable multilevel logistic regression at a p-value of <0.25. Those variables with p<0.25 were candidates and used for multivariable multilevel logistic regression. In the final model (model III), those predictors with a p-value less than 0.05 at the confidence level of 95% were considered a significant factor in emergency caesarean delivery. The adjusted odds ratio was the measure of association used to report the effect of each predictor on emergency caesarean delivery. The log of the probability of emergency cesarean delivery was modeled as follows:

$$\text{Log}(\pi_{ij}/1-\pi_{ij}) = \beta_0 + \beta_1 X_{ij} + \beta_2 Z_{ij} + u_j$$

Where ij = level I and level II units respectively
X Z = X is level I (individual level) predictors and Y is level II (community level) predictors
$\beta_1$, $\beta_2$ = the beta coefficients of each predictors
$\pi_{ij}$ = the probability of emergency cesarean delivery in the i[th] women j[th] community.
$\beta_0$ = intercept

## Ethical-consideration

Because it was secondary data, ethical consent was not applicable; rather, data was requested and authorized to be accessed from the Demographic Health Survey International (DHS). The IRB-approved procedures for DHS public-use datasets do not in any way allow respondents, households, or sample communities to be identified. There are no names of individuals or household addresses in the data files. The data will not be passed on to other researchers or a third party without the written consent of DHS International.

## Results

A total of 11,022 participants were included in the current study. Among the participants, 3339 (30.3%) of them were unemployed, and 45 (0.4%) had a history of fistula. About 8667 (76.6%) of the participants were from rural residences. Regarding their behavior, 487 (4.4%)

**Table 2. Individual and community level variables of the study population in Ethiopia using EDHS 2016 data (from 18 January to 27 June 2016).**

| Variable | Category | Weighted frequency | % |
|---|---|---|---|
| **Level I variables** | | | |
| Wealth index | Poor | 8101 | 73.5 |
| | Middle | 2045 | 18.6 |
| | Rich | 876 | 7.9 |
| Education status | No education | 6838 | 62 |
| | Primary education | 2678 | 24.3 |
| | Secondary and above | 1506 | 13.7 |
| Anemia | Yes | 6474 | 58.7 |
| | No | 4548 | 41.3 |
| Employment status | Employed | 7683 | 69.7 |
| | Not employed | 3339 | 30.3 |
| Gestational age | ≥37 weeks | 8964 | 81.3 |
| | <37 weeks | 2058 | 18.7 |
| History of fistula | Yes | 45 | 0.4 |
| | No | 10977 | 99.6 |
| Age of women | ≥35 | 4158 | 37.7 |
| | <35 | 6864 | 62.3 |
| Ever chat chewing | Yes | 431 | 3.9 |
| | No | 10591 | 96.1 |
| Types of pregnancy | Single | 10,472 | 95 |
| | Multiple | 550 | 5 |
| Birth weight | ≥3500 | 3511 | 32 |
| | <3500 | 5500 | 50 |
| | Unknown and not weighted | 2011 | 18 |
| Ever smoking | Yes | 487 | 4.4 |
| | No | 10535 | 95.6 |
| **Level II variables** | | | |
| Community media exposure | High proportion exposure | 6987 | 63.4 |
| | Low proportion exposure | 4035 | 36.6 |
| Residence | Rural | 8667 | 76.6 |
| | Urban | 2355 | 23.4 |
| Community educational status | High proportion of educated | 4233 | 38.4 |
| | Low proportion of educated | 6789 | 61.6 |

and 431 (3.9%) of them had a history of smoking and chat chewing, respectively. Furthermore, 6987 (63.4%) of the participants were from a high proportion of the media exposure community (**Table 2**).

## Prevalence of emergency cesarean delivery in Ethiopia

In the current study, the prevalence of emergency caesarean delivery was 1.2% (95% CI: 0.58, 1.78) (**Fig 2**).

## Predictors of emergency cesarean delivery

Before identifying of predictors using multilevel binary logistic regression, the clustering effect of emergency cesarean delivery using EDHS cluster (V001) was analyzed and its ICC value of

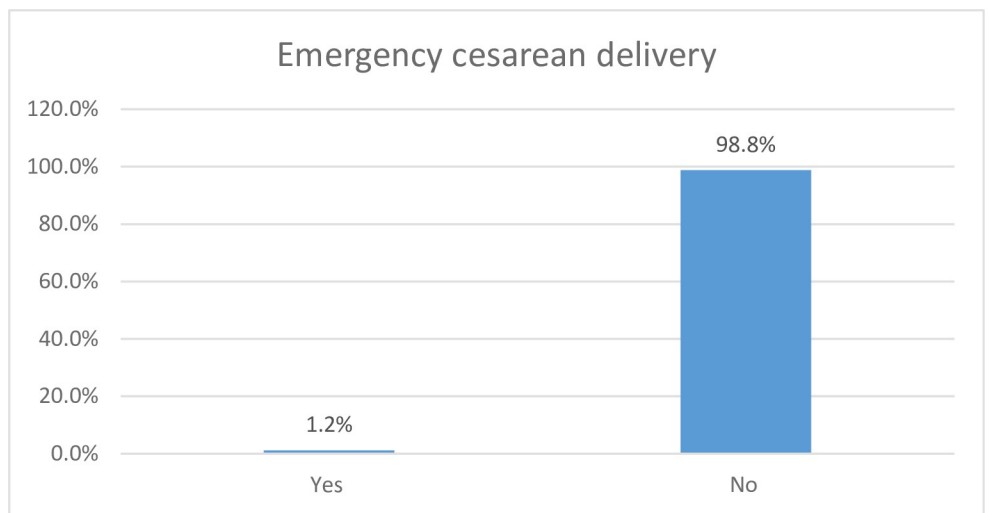

**Fig 2. The prevalence of emergency cesarean delivery in Ethiopia using EDHS-2016 (from 18 January to 27 June 2016).**

the intercept only model was 0.234 (23.4%). Four models were built. These are null model (a model with intercept or no predictors), model I (a model with individual level factors), model II (a model with community level factors) and model III (a model with mixed individual and community level factors). For each model, variance, AIC, PCV, log likelihood ratio and ICC were calculated. Then the best model was selected using the lowest value of AIC which was model III (mixed effect model). Model III (mixed effect model) had an AIC goodness of fit value 695 and PCV of 79.2% (**Table 1**). Then, bi-variable multilevel logistic regression with a p-value of <0.25 was also used to select the candidate predictors for multivariable mixed effect analysis. Predictors like wealth index, educational status, and community educational status, history of fistula, age of the women, types of pregnancy, birth weight, residence, and community media exposure were selected for multivariable mixed effect analysis. After multivariable mixed effect analysis, the age of the woman, her residence, and her history of fistulas were identified as the associated factors for emergency caesarean delivery. As such, the odds of emergency caesarean delivery among women who had a history of fistula were 7.82 times (AOR = 7.82, 95% CI: 1.59–38.4) higher than their counterparts. Women whose age was greater or equal to 35 years old were 6.98 (AOR = 6.98, 95%CI: 3.33–14.63) times more likely to give birth by emergency caesarean section than women whose age was less than 35 years old. Furthermore, women who were from rural residence were 2.23 (AOR = 2.23, 95% CI: 1.25–3.21) times more likely to give birth by emergency caesarean section than those from urban residence (**Table 3**).

## Discussion

Emergency caesarean delivery is the most important lifesaving procedure to reverse maternal and fetal deaths. But how many women give birth by emergency caesarean delivery and which maternal and fetal-related predictors are responsible for emergency caesarean delivery are not clear in Ethiopia. Therefore, in this study, an attempt has been made to assess the prevalence and predictors of emergency caesarean delivery in Ethiopia using the EDHS 2016 data set. So that, the prevalence of emergency cesarean delivery in Ethiopia among all births was 1.2% (95% CI: 0.58, 1.78). This finding was lower than a study conducted in sub-Saharan African

**Table 3. Predictors of emergency cesarean delivery in Ethiopia using EDHS-2016 (from 18 January to 27 June 2016).**

| Variables | Null model | Model I | Model II | Model III(mixed) |
|---|---|---|---|---|
| **History of fistula** | | | | |
| Yes | | 7.26 (1.46, 36.2)* | | **7.82 (1.59, 38.4)*** |
| No | | Reference | | Reference |
| **Residence** | | | | |
| Rural | | | 2.96 (1.96, 4.49)* | **2.23 (1.25, 3.21)*** |
| Urban | | | Reference | Reference |
| **Types of pregnancy** | | | | |
| Multiple | | 1.81(0.87, 3.77) | | 1.78 (0.85, 3.71) |
| Single | | Reference | | Reference |
| **Women age** | | | | |
| > = 35 years | | 6.3 (3.03, 13.2)* | | **6.98 (3.33, 14.63)*** |
| <35 years | | Reference | | Reference |
| **Wealth index** | | | | |
| Rich | | Reference | | Reference |
| Middle | | 0.94 (0.35, 2.5) | | 0.94 (0.35, 2.51) |
| Poor | | 1.3 (0.55, 3.1) | | 1.27 (0.53, 3.05) |
| **Birth weight** | | | | |
| ≥3500 gram | | 1.17(0.78, 4.12) | | 1.9 (0.84, 4.29) |
| <3500 gram | | Reference | | Reference |
| **Educational status** | | | | |
| Secondary and above | | Reference | | Reference |
| Primary education | | 1.6 (1.01, 2.7)* | | 1.21(0.32, 2.1) |
| No education | | 2.34 (1.09, 3.59)* | | 1.64 (0.84, 2.44) |
| **Community-educational status** | | | | |
| low-proportion educated | | | 1.59 (1.01, 2.51)* | 1.9 (0.84, 4.29) |
| high proportion educated | | | Reference | Reference |
| **Community media exposure** | | | | |
| Low proportion exposure | | | 1.52 (0.99, 2.34) | 1.14 (0.58, 2.24) |
| High proportion exposure | | | Reference | Reference |

Note

* statistically significant variables

countries (4.6%) [23], Nigeria (20%) [24], a study conducted among Saudi women 25% [25], 14.1% in Bangladesh [26] and 56.4% in Burkina Faso [27]. The possible reason for this discrepancy might be associated with the variation in emergency caesarean delivery service and other health service access across African countries. In Ethiopia, significant numbers of women and newborns die because of a lack of access to delivery services, including emergency delivery services, as compared to other countries. Because emergency caesarean delivery needs financial and skilled health professionals, which are limited in Ethiopia. Particularly, in the remote areas (rural areas) of Ethiopia, the demand of surgical delivery is high but yet the infrastructure to get the service is too limited and thus women in Ethiopia are suffer from low access to basic and lifesaving health services.

Regarding the predictors of emergency caesarean delivery, women who were from rural residences were more at risk of giving birth by emergency caesarean delivery than those who were from urban residences. This might be because, in the Ethiopian context, women who are from rural residences are more likely to delay reaching, decide to get delivery service at health

facilities, and have low utilization behavior of antenatal care services. Because of this, women are at risk for birth complications, which leads to an emergency caesarean delivery to save the woman and the newborn.

Older women (≥35 years) had higher odds of emergency caesarean delivery as compared to those women whose age was less than 35 years old. This finding was supported by a study conducted in in China [28], Republic of Korea [29] and study done in Yale New Haven Hospital [30]. This is because older women are more at risk for induction of labour before the onset of labour as a medical indication, and obstetric complications, for example, dysfunctional labour and intrapartum hemorrhage, are more common among older women than younger women [31] and thus, to manage this problem, emergency caesarean delivery could be the solution. Older women are also at high risk for other comorbidities during pregnancy, for example, gestational hypertension and fetal distress [32]. As a result, they may give birth by emergency caesarean section. The other possible reason might be the risk of stillbirth or prolonged and obstructed labour, which interfere with the progress of labour to the next stage and may increase the risk of emergency caesarean delivery [33, 34].

This study also showed that, women who had history of fistula were more likely to give birth by emergency caesarean delivery as compared to women who had no history of fistulas. A previous study also suggested that obstetric complications increase the risk of emergency caesarean delivery [10]. This might be associated with the fact that women with a history of fistulas may face a problem of failure in labour progress because of prolonged and obstructed labour. Obstructed and prolonged labour may be the influencing factor or the medical indication for emergency caesarean delivery.

## Strength and limitation of the study

This study used a nationally representative sample, which provides powerful evidence. But the unavailability of studies conducted on a comparable population makes it difficult to compare this study with other findings in Ethiopia. This study also used a cross-sectional research design, so it shares the limitations of this design. Additionally, because of the nature of secondary data, this study did not include additional important variables like maternal medical condition and other clinically important variables. Therefore, further research is needed to include additional clinically important predictors.

## Conclusion

As compared to the previous study, the prevalence of emergency caesarean delivery was low. Women with a history of fistula, from rural residence, and age≥ 35 years were at risk for emergency caesarean delivery. Therefore, interventions need to be encouraged to give attention to rural women whose age is ≥ 35 years and fistula reduction activities, such as avoiding early marriage, to reverse the problem. Early and accurate screening of women for emergency cesarean delivery by encouraging co-services like ANC is also recommended.

## Acknowledgments

I would like to acknowledge the international demographic health survey for its cooperation in accessing the data set. The conceptualization, data curation, formal analysis, investigation, methodology, software, supervision, visualization, and writing of the original draft were performed by the principal investigator.

## Author Contributions

**Conceptualization:** Muluken Chanie Agimas.

**Data curation:** Muluken Chanie Agimas.

**Formal analysis:** Muluken Chanie Agimas.

**Software:** Muluken Chanie Agimas.

**Supervision:** Muluken Chanie Agimas.

**Validation:** Muluken Chanie Agimas.

**Visualization:** Muluken Chanie Agimas.

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
