## [Decision Letter · Decision Letter 0]

8 Jan 2024

PONE-D-23-35082Emergency cesarean delivery and its predictors in Ethiopia using further analysis of EDHS 2016 data: A mixed effect modelPLOS ONE

Dear Dr. chanie,

Thank you for submitting your manuscript to PLOS ONE. After careful consideration, we feel that it has merit but does not fully meet PLOS ONE’s publication criteria as it currently stands. Therefore, we invite you to submit a revised version of the manuscript that addresses the points raised during the review process.

**ACADEMIC EDITOR: **please address comments given point by point

We look forward to receiving your revised manuscript.

Kind regards,

Temesgen Tilahun

Academic Editor

PLOS ONE

Journal Requirements:

Additional Editor Comments:

Dear authors,

please address the following.

1. put the sampling procedure in flow diagram

2. include limitations

3. look at your findings again and include additional recommendations

Reviewers' comments:

Reviewer's Responses to Questions

**Comments to the Author**

1. Is the manuscript technically sound, and do the data support the conclusions?

Reviewer #1: Yes

Reviewer #2: No

Reviewer #3: Yes

Reviewer #4: Partly

2. Has the statistical analysis been performed appropriately and rigorously? 

Reviewer #1: Yes

Reviewer #2: Yes

Reviewer #3: Yes

Reviewer #4: N/A

3. Have the authors made all data underlying the findings in their manuscript fully available?

Reviewer #1: Yes

Reviewer #2: No

Reviewer #3: Yes

Reviewer #4: Yes

4. Is the manuscript presented in an intelligible fashion and written in standard English?

Reviewer #1: Yes

Reviewer #2: No

Reviewer #3: Yes

Reviewer #4: Yes

5. Review Comments to the Author

Reviewer #1: General Comment: The content is important to the field and the area is researchable. There is also a significant maternal health services provision, thus improving health of maternity services.

Abstract: The abstract reflect aspects of the study clearly: background, objectives, methods, results and conclusions, but in the methods part selecting method of model( what means “the lowest Akaike information criteria” the author should be make it clear explain). And author missed to leave educational status set as “significant predictors of emergency cesarean delivery” because, it wasn’t show association under AOR or multivariable analysis

Introduction: The study rationale satisfactorily described but as gaps: limited to describe , gaps and plan or purpose of study details and the author should address that it needs highlight revise

Methodology: The study design appropriate and adequate for the plan of study, The sample size seems an appropriate but inadequate justified in case of set this value(11,022 included in analysis),the author should clarify the gaps and also needs highlight revise sampling technique. The author also missed to identify their number and professional of data collators, The author missed to state the exclusion criteria’s that what action takes place in study during study period, Statistical analysis preferred analysis version is best and it is better to state the value of goodness of fit of the model.

Result: The findings were presented logically with appropriate displays but, the author missed that the statement of bi-variable multilevel logistic regression at a p-value of <0.25 should be narrated under methods and also two lines sentences under the sub topic “Predictors of emergency cesarean delivery” should be stated under methods sub topic statistical analysis.

In the title of tables and figure should be also incorporate study period of Ethiopian demographic health survey (EDHS)2016 data.

Discussion: The key findings were stated, almost it seems well constructed and the findings also discussed in the light of previous evidence but it shows the limitation of comparing findings with developing countries like African especially Sub-Saharan countries.

Conclusion: The results justify the conclusion is fair.

Bibliography/References: The references appropriate and relevant, except a few references such as ref. NO9, 10 and 12 are later that needs replace with recent.

Others: Overall the paper presented logically and as a result, an author is advised to make some revisions then the paper will be become the best.

Decision: Minor Revision with an attention.

Reviewer #2: Q1

1. Is the manuscript technically sound, and do the data support the conclusions?

No, The manuscript, while addressing a good topic, appears to lack technical soundness. For instance, a statement in the manuscript mentions, "Specifically, in Ethiopia, the rate of all forms of cesarean delivery is from 8-37% (5, 13). But nothing is known about emergency cesarean delivery," which may not be accurate. This claim contradicts information found in a relevant paper accessible through this link: https://bmcpregnancychildbirth.biomedcentral.com/articles/10.1186/s12884-021-04266-7. Additionally, a comparison with another paper (file:///C:/Users/belac/Downloads/journal.pone.0282951%20) reveals significant differences in the author's contributions to the study.

Q3

*3. Have the authors made all data underlying the findings in their manuscript fully available?

The author should mention in details restrictions on publicly sharing data—e.g. participant privacy or use of data from a third party—those must be specified.

Q4

4. Is the manuscript presented in an intelligible fashion and written in standard English?

strongly advise the author to edit the grammer.

Overall, I believe this manuscript may not meet the publication criteria set by PLOS ONE based on the aforementioned concerns I have raised.

Reviewer #3: Abstract

Methods

1. Was the entire study sample of the EDHS 2016 considered for the study?

2. If not how, this sample (i.e., 110222) was selected?

3. What makes this study Multilevel?

4. “...The best model was selected based on the lowest Akaike information criteria..." So, what is the best model selected explain it?

Results

5. Are these micro-level or Macro-level variables?

6. What about the intraclass correlation (ICC) report? Have checked whether or not there are clustering of the cesarean section?

Main Body

Introduction

7. Line 51-52, "... According to the world health organization (WHO) recommendation, the cesarean delivery should be from 5-15%..." is this information about elective or emergency cesarean section?

8. What is the expected magnitude of Emergency cesarean delivery, according to WHO?

9. Line 65-67, if numerous studies were conducted, why you failed to mention a magnitude of cesarean section in Ethiopia?

Methods

10. Line 75-77, better to update the regions of Ethiopia based on the recent classification?

11. Line 78, "... world speedometer..." Is it appropriate term? Worldometer? Is the report of worldometer valid and acceptable?

12. Line 81-83, were elective women for cesarean delivery, who undergone cesarean section two or more times eligible for this study?

13. Line 98, the sampling procedure is not clear? Is to mean multistage random sampling or what?

14. Line 99, it is not clear how a sample size of the study was determined?

15. Line 120, It is better if the best model selected was reported instead.

16. Line 126, before using the multilevel model, It is must to elaborate the clustering variable either in the operational definition or elsewhere?

17. Line 128, it is not clear which variables are considered as "X" or "Z".

Result

18. Line 147, before rushing to identification of predictors using multilevel binary logistic regression, testing of the appropriateness of the model required. So, the author must clarify the values of ICC, PCV, and AIC, that reported in the table 1. Additionally, it is not clear what model I, Model II, and Model III are? On the other way, the clustering variable is not clear to apply the model?

19. Line 153, what is the term "... After all..." means? Is it to mean that after doing Crude analysis (bivariate binary logistic regression)? It needs more detail explanations?

Discussion

20. Line 169-172, in spite of what you justified, do you think it is fair to compare Japan with Ethiopia?

21. Add more comparison and discuss it as this result is not well discussed?

22. In general your discussion is comparing incomparable population and results?

23. Line 175-176, how do you see comparing all type of cesarean section to emergency cesarean section (your study)?

24. What is the strengthen of this study?

Reviewer #4: Overall the manuscript is not bad in short of listed below comments

>Topic is not " SMART" and lacks novelty =revise it

>Data source is secondary and not current which might affects reliability and accuracy of the result

>Try to clarify your research methods and materials and significance of your study

6. PLOS authors have the option to publish the peer review history of their article (what does this mean?). If published, this will include your full peer review and any attached files.

Reviewer #1: **Yes: **Girma Worku Obsie

Reviewer #2: No

Reviewer #3: No

Reviewer #4: **Yes: **Lammii Gonfaa Dinagde

---

## [Author Response · Author response to Decision Letter 0]

25 Jan 2024

Dear reviewers and editor the response for the comment are attached with "response to reviewer document"

---

## [Editor Report · Decision Letter 1]

29 Feb 2024

Prevalence of emergency caesarean delivery and its predictors among women who give birth in Ethiopia using further analysis of EDHS 2016 data: A mixed effect model

PONE-D-23-35082R1

Dear Author ,

We’re pleased to inform you that your manuscript has been judged scientifically suitable for publication and will be formally accepted for publication once it meets all outstanding technical requirements.

Kind regards,

Temesgen Tilahun

Academic Editor

PLOS ONE
---

## [Editor Report · Acceptance letter]

21 Mar 2024

PONE-D-23-35082R1 

PLOS ONE

Dear Dr. Agimas, 

I'm pleased to inform you that your manuscript has been deemed suitable for publication in PLOS ONE. Congratulations! Your manuscript is now being handed over to our production team.

Kind regards, 

on behalf of

Dr. Temesgen Tilahun 

Academic Editor

PLOS ONE